# Pathogenic Effects of Mineralocorticoid Pathway Activation in Retinal Pigment Epithelium

**DOI:** 10.3390/ijms22179618

**Published:** 2021-09-05

**Authors:** Jérémie Canonica, Min Zhao, Tatiana Favez, Emmanuelle Gelizé, Laurent Jonet, Laura Kowalczuk, Justine Guegan, Damien Le Menuet, Say Viengchareun, Marc Lombès, Eric Pussard, Yvan Arsenijevic, Francine Behar-Cohen

**Affiliations:** 1Centre de Recherche des Cordeliers, Sorbonne Université, Université de Paris, Inserm, From Physiopathology of Retinal Diseases to Clinical Advances, 15 rue de l’Ecole de Médecine, 75006 Paris, France; jerem.canonica@gmail.com (J.C.); min.zhao@inserm.fr (M.Z.); emmanuelle.gelize@gmail.com (E.G.); laurent.jonet@crc.jussieu.fr (L.J.); 2Department of Ophthalmology, Jules Gonin Eye Hospital, Fondation Asile des Aveugles, University of Lausanne, 1004 Lausanne, Switzerland; tatiana.favez@epfl.ch (T.F.); laura.kowalczuk@epfl.ch (L.K.); Yvan.Arsenijevic@fa2.ch (Y.A.); 3Institut du Cerveau, ICM, iCONICS, Hôpital de la Pitié-Salpêtrière, 75013 Paris, France; justine.guegan@icm-institute.org; 4Physiologie et Physiopathologie Endocriniennes, Université Paris-Saclay, Inserm, 94276 Le Kremlin-Bicêtre, France; damien.le-menuet@u-psud.fr (D.L.M.); say.viengchareun@universite-paris-saclay.fr (S.V.); marc.lombes@universite-paris-saclay.fr (M.L.); eric.pussard@aphp.fr (E.P.); 5Assistance Publique—Hôpitaux de Paris, Hôpital Bicêtre, Service de Génétique Moléculaire et d’Hormonologie, 94276 Le Kremlin Bicêtre, France; 6Assistance Publique—Hôpitaux de Paris, Hôpital Cochin Ophthalmopole, 75014 Paris, France

**Keywords:** retina, eye, corticoids, retinal pigment epithelium, mineralocorticoid, transcriptional regulation

## Abstract

Glucocorticoids are amongst the most used drugs to treat retinal diseases of various origins. Yet, the transcriptional regulations induced by glucocorticoid receptor (GR) and mineralocorticoid receptor (MR) activation in retinal pigment epithelium cells (RPE) that form the outer blood–retina barrier are unknown. Levels of endogenous corticoids, ligands for MR and GR, were measured in human ocular media. Human RPE cells derived from induced pluripotent stem cells (iRPE) were used to analyze the pan-transcriptional regulations induced by aldosterone—an MR-specific agonist, or cortisol or cortisol + RU486—a GR antagonist. The retinal phenotype of transgenic mice that overexpress the human MR (P1.hMR) was analyzed. In the human eye, the main ligand for GR and MR is cortisol. The iRPE cells express functional GR and MR. The subset of genes regulated by aldosterone and by cortisol + RU-486, and not by cortisol alone, mimics an imbalance toward MR activation. They are involved in extracellular matrix remodeling (*CNN1*, *MGP*, *AMTN)*, epithelial–mesenchymal transition, RPE cell proliferation and migration (*ITGB3*, *PLAUR* and *FOSL1*) and immune balance (*TNFSF18* and *PTX3*). The P1.hMR mice showed choroidal vasodilation, focal alteration of the RPE/choroid interface and migration of RPE cells together with RPE barrier function alteration, similar to human retinal diseases within the pachychoroid spectrum. RPE is a corticosteroid-sensitive epithelium. MR pathway activation in the RPE regulates genes involved in barrier function, extracellular matrix, neural regulation and epithelial differentiation, which could contribute to retinal pathology.

## 1. Introduction

Retinal diseases are the most frequent causes of blindness in developed countries [1], and one of the major causes of vision loss is macular edema [2] that complicates retinal barriers’ breakdown. Blood–retina barriers are formed in the inner retina by endothelial cells of retinal vessels and in the outer retina by the retinal pigment epithelium (RPE) [2]. The RPE is a highly polarized monolayer of tight-junction cells that, besides its barrier function, ensures retinoid metabolism, photoreceptor outer segment phagocytosis, immune surveillance and ion and water transports, required for retinal homeostasis [2,3,4].

Intraocular glucocorticoids (GCs) reduce macular edema of various origins [5,6], but exogenous and even endogenous corticoids may induce paradoxical pro-edematous effects in pachychoroid-spectrum diseases including central serous chorioretinopathy (CSCR) [7], witnessing the complexity of their action on the retina. The role of corticoids in retinal homeostasis and in diseases remains incompletely understood and underestimated. In the rat retina, neurosteroids produced by steroidogenesis in rat retinal ganglion cells [8] protected retinal neurons against ischemia through binding to the sigma-1 receptor [9], which regulates calcium, ion channels and oxidative stress [10]. In a human RPE cell line (ARPE-19), 17β-estradiol and dehydroepiandrosterone-sulfate protected against oxidative-stress-induced DNA damages [11], demonstrating the crucial role of neurosteroids in retina and RPE maintenance. In the same RPE cell line, key enzymes participating in steroid synthesis were identified and transformation of progesterone into cortisol was found in culture media, suggesting that an equivalent of hypothalamic–pituitary–adrenal axis in adult retinal pigment epithelium may exist [12]. GCs act through binding to the glucocorticoid (GR) and the mineralocorticoid receptor (MR), both expressed in various cells in the retina in rodents and in humans, including the RPE [13,14,15,16]. Evidence from animal and cellular models converges to indicate that in the retina, as in other non-classical MR-sensitive tissues, MR pathway overactivation is pathogenic [15,16,17,18,19,20,21] and causes pathogenic features similar to CSCR, a disease that develops due to focal disruption of the RPE barrier function [19]. Indeed, GCs were shown to regulate the RPE functions [22,23,24] and aldosterone to cause RPE/choroid pathology in vivo [21]. However, the transcriptional regulations induced specifically in the RPE by MR and/or GR activation are unknown.

The aim of this study was to analyze the pan-transcriptional regulations induced in human induced pluripotent stem cell (hiPSC)-derived RPE (iRPE) cells by corticoids with specific inhibitors of their receptors to decipher specific genes regulated by MR or GR activation. P1.hMR transgenic mice that express the human MR were further used to analyze the consequence of MR overactivation on the retina.

## 2. Results

### 2.1. RPE Cells Differentiated from hiPSC Express RPE Characteristic Marker Genes and Exhibit High Transepithelial Resistance

We used a modified protocol described by Singh, R. and colleagues [25] to differentiate and expand in 60 days hiPSC into pure iRPE cells [26] (Figure 1a) that display many features of a native RPE [27]. Experiments were performed on serially expanded P3 iRPE cells at day 42.

The iRPE cells cultured on a transwell grew in monolayers of polygonal pigmented cells (Figure 1b,c) and formed a functional barrier as shown by occludin and ZO-1 immunostaining (Figure 1d) and by a high transepithelial resistance (TER) (306.60 ± 81.49 Ω cm^2^) [25] (Figure 1e). Most of RPE characteristic marker genes involved in the visual cycle, phagocytosis, pigment synthesis and ion channel transport were expressed in iRPE cells, human fetal RPE (hfRPE) cells and in post-mortem human RPE (phRPE) controls (Figure 1f,g). Expression of RPE65, RLBP1, MERTK and BEST1 was also demonstrated at the protein level by immunolocalization in iRPE (Figure 1h). These results indicate that P3 iRPE cells possessed morphological and functional characteristics and gene and protein expression profiles similar to human RPE.

### 2.2. Mineralocorticoid and Glucocorticoid Receptors Are Stably Expressed and Are Functional in iRPE Cells

No trace of any exogenous corticosteroids was detected in the cell culture media added to iRPE (Appendix A), and the human RPE did not synthesize or secrete glucocorticoid or mineralocorticoid hormones, contrarily to what was previously reported [12]. In addition, while expression of *HSD11B1* was detected in a fresh phRPE, it was not detected in either hfRPE or in iRPE cells (Figure 2a,b). *HSD11B2* could not be detected in either human RPE samples tested or in iRPE cells, suggesting that in our cellular model, cortisol added in the culture media would not be metabolized into inactive forms.

Physiologic corticoid levels were measured in the aqueous humor of patients operated for cataract surgery (8 males, 68 ± 8.7 years old, without any retinal or systemic pathology) and in the vitreous of patients operated for epiretinal membrane (12 males, 64 ± 5 years old, without any other retinal or systemic pathology). The mean cortisol (F) concentration was 3.2 ± 1.4 ng/mL in aqueous humor and 2.94 ±1.2 ng/mL in the vitreous fluid; the mean cortisone (E) concentration was 0.7 ± 0.4 ng/mL in aqueous humor and 1.23 ± 0.6 ng/mL in the vitreous; and the E/F ratio was significantly higher in the vitreous than in the aqueous humor (0.21 ± 0.09 versus 0.42 ± 0.11, *p* < 0.0001) (Figure 2c–e), indicating that 11βHSD2 enzyme activity is higher in the posterior segment of the human eye. The mineralocorticoid hormones aldosterone, 11-deoxycorticosterone and 18-hydroxycorticosterone were not measurable in iRPE cells, human RPE/choroid explant culture media or in human aqueous and vitreous humors, indicating that in physiological conditions cortisol is the main natural ligand of MR in the retina.

Similar *NR3C2* (encodes MR) and *NR3C1* (encodes GR) expression was found in iRPE cell, hfRPE and phRPE control samples (Figure 2a,b). *NR3C2* (Figure 2f) and *NR3C1* (Figure 2g) expression was not modified by various concentrations of aldosterone or cortisol, demonstrating that iRPE stably express MR and GR. Both aldosterone and cortisol induced the translocation of MR (Figure 3a,b) and GR (Figure 3d,e) from the cytoplasm to the nucleus, as shown by immunofluorescence staining and Western blot of proteins obtained from cytoplasmic and nuclear fractions. When compared to unliganded MR, both aldosterone and cortisol increased the apparent molecular weight of MR detected on Western blots (Figure 3b), suggesting receptor phosphorylation [28,29]. Mouse cortical collecting duct (mCCD) cells with known functional endogenous MR expression served as MR positive control [30] (Figure 3b). In addition, both aldosterone (Figure 3c) and cortisol (Figure 3f) up-regulated the expression of known corticosteroid target genes, such as FKBP prolyl isomerase 5 (*FKBP5*), glucocorticoid-induced leucine zipper protein (*GILZ*), period circadian regulator 1 (*PER1*), sodium channel epithelial 1 alpha subunit (*SCNN1A*) and serum/glucocorticoid-regulated kinase 1 (*SGK1*). Gene expression of neutrophil gelatinase-associated lipocalin (NGAL encoded by *LCN2*), a mineralocorticoid target in the cardiovascular system [31], was up-regulated only following cortisol treatment (Figure 3f). These results show that the RPE is a non-classical mineralocorticoid epithelial cell layer deprived of the cortisol-inactivating enzyme 11βHSD2.

### 2.3. RNA-Seq Transcriptome Analysis Identified Corticosteroid-Regulated Genes in iRPE Cells

RNA-Seq data analysis identified 90 up- and 122 down-regulated candidate target genes for MR activation in aldosterone-treated iRPE cells when compared to untreated cells (Figure 4a,b and Appendix A). Comparative analysis between transcriptome profiles of cortisol-treated iRPE cells and untreated cells predicted 99 up- and 41 down-regulated genes to be differentially expressed by cortisol (Figure 4c,d and Appendix A). Cortisol and RU-486 co-treatment in iRPE cells resulted in 123 up- and 137 down-regulated genes compared to untreated cells (Figure 4e,f and Appendix A). Several genes were regulated by corticosteroids in a similar manner in all three experimental treatments. Indeed, C-X-C motif chemokine receptor 4 (*CXCR4*), dual-specificity phosphatase 6 (*DUSP6*), EPH receptor A8 (*EPHA8*), inhibitory synaptic factor family member 2B (*FAM196B*), *FKBP5*, hydrogen voltage-gated channel 1 (*HVCN1*), *PER1*, prostaglandin E receptor 2 (*PTGER2*), S100 calcium-binding protein A2 (*S100A2*), *SCNN1A*, serpin family A member 3 (*SERPINA3*) and transmembrane protein 72 (*TMEM72*) expression was increased in iRPE cells treated with either aldosterone, cortisol or cortisol + RU-486 (Figure 4g,h). Common down-regulated genes were calponin 1 (*CNN1*), gap junction protein delta-2 (*GJD2*), oligodendrocyte myelin glycoprotein (*OMG*) and somatostatin (*SST1*) (Figure 4g,i).

Interestingly, expression of some genes (10 up-regulated genes and 3 down-regulated genes) was comparable in the aldosterone and cortisol + RU-486 experimental conditions, representing genes resulting mostly from MR pathway activation (Figure 4g–i).

### 2.4. Validation of RNA-Seq-Predicted Corticosteroid-Regulated Genes

Some of the genes predicted by the RNA-Seq data to be differentially expressed were tested by RT-qPCR analysis for validation on samples distinct from those used for the RNA-Seq analyses (Figure 5 and Scheme 1). Good correlations were obtained between RNA-Seq and RT-qPCR results (Figure 5a–c) following aldosterone or cortisol treatment for *SERPINA3* (Figure 5d,i), *SCNN1A* (Figure 5e,j), *S100A2* (Figure 5f,k), *PTGER2* (Figure 5g,l) and *PER1* (Figure 5h,m). Up-regulation of these genes by the aldosterone was MR-dependent, as shown by the blocking effect of the MR antagonist spironolactone. Cortisol induced the regulation of selected genes that were both MR- and GR-dependent such as *SERPINA3* (Figure 5i), *SCNN1A* (Figure 5j), *PTGER2* (Figure 5l) and *PER1* (Figure 5m). Interestingly, the potency to counteract the effect of the glucocorticoid hormone differed between the MR and the GR antagonist and was specific to the targeted genes. Indeed, up-regulation in *SCNN1A* and *PER1* gene expression by cortisol was mostly reduced by the GR antagonist RU-486 and to a lesser extent by the MR antagonist spironolactone. On the other hand, RU-486 had a lesser effect than spironolactone to block the increased expression of *SERPINA3*, *S100A2* and *PTGER2*. Furthermore, aldosterone or cortisol increased *SERPINA3* (Figure 5n), *SCNN1A* (Figure 5o), *S100A2* (Figure 5p), *PTGER2* (Figure 5q) and *PER1* (Figure 5r) expression in a dose-dependent manner in iRPE cells. Other genes (*ABCC3*, *HMGA1*, *PLAUR* and *PTX3*) were validated by RT-qPCR (Scheme 1).

### 2.5. αENaC and S100A2 Protein Expression Was Increased by Aldosterone and Cortisol in iRPE Cells

Immunostaining results show that SCNN1A (Figure 6a) and S100A2 (Figure 6b) protein expression was increased in iRPE cells following either aldosterone or cortisol treatments. Interestingly, SCNN1A was predominantly expressed on the apical side of iRPE cells as highlighted by the white arrowheads (Figure 6a). Furthermore, Western blot analysis demonstrated that both aldosterone (Figure 7a,b) and cortisol (Figure 7c,d) up-regulated the relative protein expression of S100A2 and PTGER2.

### 2.6. Corticosteroids Markedly Modified the Expression Pattern of Extracellular Proteins in iRPE Cells

The Reactome pathway analysis revealed that extracellular matrix (ECM) proteoglycans, integrin cell surface interactions and degradation of the ECM terms were enriched in the ECM organization category (Figure 8a–c, Appendix A). Particularly, aldosterone down-regulated *FMOD*, *BGN* and *COL2A1* genes, while cortisol down-regulated *SPP1*, *ITGA11* and *COL1A1* genes. Moreover, activation of the MR pathway down-regulated the expression of small leucine-rich proteoglycans such as biglycan, lumican and fibromodulin, known to be highly expressed in the RPE, Bruch membrane, choroid and sclera [32]. Cortisol and RU-486 co-treatment regulated numerous pathways suggesting that the activation of MR by cortisol has more biological consequences than activation of MR by aldosterone. Indeed, several pathways involved in the control of cell proliferation (cell cycle category) were down-regulated (Figure 8c, Appendix A). In the homeostasis category, the basigin interactions pathway known to be associated with the synthesis of several matrix metalloproteinases, angiogenesis via stimulation of VEGF production and glucose uptake was up-regulated. Other categories in the Reactome pathway analysis to be significantly represented in corticosteroid-treated iRPE cells were genes involved in metabolism of proteins, signal transduction and metabolism. 

With regard to the biological process GO analysis, cell communication, anatomical structure development, cell adhesion, cellular metabolic process, cell death, regulation of biological quality, cellular component organization or biogenesis and cell cycle were the most represented categories enriched in corticosteroid-treated iRPE cells (Figure 9a–c, Appendix A). Regarding the molecular function GO, the most represented categories in the aldosterone, cortisol and cortisol + RU-486 experimental conditions included signaling receptor activity, protein binding, transmembrane transporter activity, ECM binding and oxidoreductase activity (Scheme 2, Appendix A). Finally, concerning the cellular component GO, categories involved at the membrane, cell junction, extracellular region, cell periphery and intracellular anatomical structure level were represented in corticosteroid-treated iRPE cells, which comprised enrichments of integrin complex, focal adhesion, proteinaceous ECM, ECM and microvillus as terms of interest (Scheme 3, Appendix A).

### 2.7. Retinal Phenotype of Mice Overexpressing the Human MR Showed Abnormalities at the Level of the RPE and the Choroidal Vasculature

While no difference in murine *Nr3c2* (Figure 10a) and *Nr3c1* (Figure 10b) expression was observed between P1.hMR and control animals, the human *NR3C2* transgene was highly expressed in the RPE/choroid complex of P1hMR mice (Figure 10c). As expected, no expression of the transgene was detected in wild-type control mice. In 6-weeks-old mice, in vivo spectral-domain optical coherence tomography (SD-OCT) imaging showed a normal structure of the inner retina in P1.hMR mice (Figure 10e,f). Nevertheless, an increased thickness of the choroid (Figure 10e,f, insets 3, 4 and 6) and hyperreflective and dense irregular areas at the level of the RPE (Figure 10e,f, insets 3–5) were observed. Epithelial and choroidal abnormal features of the outer retinal structure were absent in control animals (Figure 10d, insets 1 and 2).

Histological section analysis confirmed that as compared to wild-type mice (Figure 11a and inset 1), the choroidal thickness in P1.hMR mice was increased, as shown by enlarged choroidal vessels, inducing RPE cell displacement and focal retinal detachment (Figure 11b,c,e insets 2, 3 and 5). Moreover, RPE cells appeared swollen with intracellular vacuole accumulation (Figure 11b–e, insets 2–5). In addition, inner and outer segments of the photoreceptor were elongated (Figure 11b,c,e, insets 2, 3 and 5). The surface area of the choroid (2.367 ± 0.719 vs. 1.402 ± 0.098 mm^2^) (Figure 11f), photoreceptor segments (4.711 ± 1.141 vs. 3.661 ± 0.414 mm^2^) (Figure 11g) and RPE thickness (4.3 ± 0.22 vs. 3.6 ± 0.6 µm) (Figure 11h) were significantly increased in P1.hMR mice as compared to control eyes. Whist albumin localized only into choroidal vessels and faintly in RPE cells in wild-type mice, it showed accumulation within and between RPE cells, in the choroidal stroma and in the photoreceptor segments in P1.hMR transgenic mice (Scheme 4). Overexpression of the human MR encoded by the *NR3C2* gene in the retina, without any other exogenous stress, induced progressive pathology, mostly observed at the level of the RPE and the choroid.

## 3. Discussion

The iRPE model stably expresses functional MR and GR and expresses known target genes upon corticoid exposure. Although the lowest aldosterone dose required to induce the expression of target genes was higher than expected, specificity of MR activation was demonstrated by the inhibitory effect of the MR antagonist spironolactone and by the dose-dependent effect of aldosterone on the expression of several genes such as *SERPINA3*, *SCNN1A*, *S100A2*, *PTGER2* and *PER1.* The model is thus appropriate to study GR and MR transcriptomic regulations using specific ligands. In humans, aldosterone was below detectable levels in ocular media, suggesting control of its entry in the eye, as it was described in the brain [33,34]. Cortisol should thus be the preferential ligands for GR and MR in RPE cells in vivo although it cannot be excluded that aldosterone could stimulate RPE cells from their basolateral.

In cortisol-treated cells, the number of significantly up-regulated genes was higher than the number of down-regulated genes. The contrary was observed in aldosterone- and cortisol + RU486-treated cells, indicating that MR activation represses rather than induces gene expression.

A subset of genes, regulated in a similar manner in all three experimental treatments, results from both GR and MR activation. Amongst those genes, *CXCR4* and *PTGER2* encode receptors that upon activation by their specific ligand SDF-1 and PGE2 can alter RPE functions. SDF-1 binding to CXCR4 induces RPE migration and contributes to retinal inflammation and choroidal neovascularization [35] whilst PGE2 induces RPE contraction and disruption of its barrier function [36]. In addition, PGE2 inhibits the ingestion of rod outer segments by RPE cells [37]. Up-regulation of *CXCR4* and *PTGER2* expression could alter RPE functions in a pro-inflammatory environment when their ligands are produced. *HVCN1* encodes a voltage-gated proton channel that regulates the decrease in pH and reactive oxygen species burst induced by phagocytosis [38,39], a rhythmic process, regulated by clock genes in RPE cells [40]. Interestingly both aldosterone and cortisol up-regulated the expression of the clock gene *PER1*, in a dose-dependent manner. Cortisol could contribute to the circadian regulation of photoreceptors’ outer segment recycling. The role of S100A2, recognized as a bad prognosis biomarker in cancer [41], is not known in the eye. S100A2 was shown to inhibit transforming growth factor-β1 (TGF-β1)-induced epithelial–mesenchymal transition (EMT) in lung epithelial cells [42] and, upon oxidative stress, to translocate into the cytoplasm and delay keratinocytes’ cell death [43]. In iRPE cells, while cortisol favored the nuclear retention of the protein, aldosterone, and cortisol + RU486 rather induced its cytoplasmic translocation, suggesting differential effects of GR or MR activation on S100A2 activities.

The Reactome and GO analysis showed the important role of GR and MR activation in RPE cell differentiation. *SERPINA3* encodes alpha-1 antichymotrypsin (ACT), which inhibits the activity of the serine proteases cathepsin G and chymase, secreted from mast cells, that are abundant in the choroid. Chymase activates matrix metalloproteinase (MMP)-9 and TGF-β that are essential for Bruch membrane maintenance and for extracellular matrix (ECM) [44]. All these genes encode proteins that intervene in RPE differentiation and adhesion and in RPE phagocytosis. Finally, *SCNN1A*, encoding alpha-ENaC- channel, expressed in RPE cells [45] is up-regulated by cortisol and by aldosterone as previously shown [16,21]. Immunohistochemistry of alpha-ENaC- performed on transversal sections of iRPE cells confirmed that aldosterone and cortisol increased its expression particularly toward the apical side.

Genes that were down-regulated by three corticosteroid treatments are calponin 1 (*CNN1*), somatostatin (*SST*), gap junction protein delta-2 (*GJD2)*, and oligodendrocyte myelin glycoprotein (*O**MG*). Calponin’s role in RPE cells is unknown. Somatostatin, whose receptors are expressed in RPE cells [46], protects its barrier function [47] and regulates NO production [48]. Down-regulation of *SST* and *GJD2* might thus fragilize the RPE barrier. Finally, down-regulation of *OMG* in RPE cells, which encodes oligodendrocyte–myelin glycoprotein, could indicate a role for the RPE in the maintenance of choroidal nerves, which control the choroidal blood flow [49] through a secreted isoform of OMG [50].

The subset of genes that are up- or down-regulated by aldosterone and cortisol + RU-486, and not by cortisol alone, mimics the effect of an imbalance toward MR overactivation in human RPE cells. One group of genes encodes proteins involved in ECM remodeling, epithelial–mesenchymal transition (EMT) and RPE cell proliferation and migration, such as *ITGB3* encoding the integrin subunit beta-3, *PLAUR* encoding the urokinase plasminogen activator receptor (uPAR) and *FOSL1* encoding the FOS like 1 AP-1 transcription factor subunit protein. Interestingly, FOSL1 controls the expression of *PLAUR* and *ITGB3* [51]. The uPAR is expressed at the basolateral membrane of RPE cells [52], and activation of uPA-uPAR pathways favors choroidal neovascularization [53] and induces RPE EMT and proliferation [54]. In addition, the uPAR has been shown to interact with the integrin subunit beta-3 expressed on RPE cells [55,56], inducing extracellular proteolysis by enhancing cell surface plasminogen activation. Three other down-regulated genes (*CNN1*, *MGP* and *AMTN*) encode proteins involved in the rigidity of the ECM. *MGP* encodes the matrix glia protein, a calcium-binding ECM protein with the strongest inhibitory effect on vascular calcification. It is expressed in the trabecular meshwork and the anterior sclera, the suprachoroidal space and around the optic nerve [57]. Deficiency in MGP induces vascular calcification and cerebral arteriovenous malformations [58]. Down-regulation of MGP could favor calcification and rigidity of choroidal vessels.

*TNFSF18* and *PTX3*, up-regulated by MR activation, encode proteins involved in immune and inflammatory balance. *TNFSF18*, encoding the glucocorticoid-induced TNF-related ligand 3 (GITR3) in RPE cells [59], abrogates RPE-mediated immunosuppression of CD3^+^ T-cells which is considered as a possible mechanism for ocular immune privilege [60]. Its overexpression could thus alter the subretinal immune suppression. On the other hand, pentraxin 3 (*PTX3*) overexpression binds factor H and prevents oxidation-mediated activation of inflammasome in RPE cells [61].

Several genes encoding proteins involved in RPE metabolism are down-regulated. *ABCC3*, which encodes the multidrug resistance protein 3 (MRP3) in RPE cells [62], regulates the efflux of glucuronide steroids including estradiol and testosterone. And GDNF family receptor alpha 2 (encoded by *GFRA2)* could be involved in decreased photoreceptor (OS) phagocytosis [63].

Altogether, these results indicate that MR overactivation in RPE cells by endogenous ligands could favor pathogenic features. This hypothesis was further explored in mice overexpressing the human MR. At 9 months, P1.hMR mice showed choroidal vasodilation, as observed previously in rodents treated with acute aldosterone ocular injection [14], and elongation of photoreceptors OS, reflecting a decrease in RPE phagocytosis, focal alteration of the RPE/choroid interface and migration of RPE cells. In addition, albumin staining showed an increase in transport in the RPE cells and possible leakage through the RPE barriers, reflecting alteration of the RPE barrier. Interestingly, no major change was observed in the neural retina of these mice. The retinal phenotype of P1.hMR mice mimics human pachychoroid pigment epitheliopathy [64]. Interestingly, numbers of genes and pathways, identified in the iRPE cell model were also regulated in the rat RPE/choroid complex by acute aldosterone injections [21].

In conclusion, this study shows that the RPE is a corticosteroid-sensitive epithelium and that specific MR pathway overactivation by endogenous corticoids regulates genes involved in RPE phagocytosis of OS, inflammation and immunity, barrier function, ECM remodeling, ion transports and EMT transition. The P1.hMR transgenic mouse model confirmed the pathogenic role of MR overactivation in the retina and its possible link with the pachychoroid-disease spectrum. Further studies should confirm these findings and molecular targets on other RPE cell lines.

## 4. Material and Methods

### 4.1. Human iPSC Differentiation into RPE Cells

Human iPSCs obtained from a healthy donor (Dr. David M. Gamm, Department of Ophthalmology and Visual Sciences, University of Wisconsin–Madison, United States) were expanded and differentiated into iRPE as described previously [26] (Figure 1a). TER values were measured weekly with an epithelial voltohmmeter device (EVOM2, World Precision Instruments, Friedberg, Germany) using standard chopstick fixed double electrodes.

### 4.2. Corticosteroid Treatments of iRPE Cultured Cells

Cells were seeded at P3 in cell culture plastic dishes or transwell filters. On day 35, one week prior to corticosteroid treatments, RDMw/oA medium was removed and iRPE cells were incubated in experimental corticosteroid-free medium (DMEM, high glucose, HEPES, no phenol red (Thermo Fisher Scientific, Saint Aubin, France)); 10% fetal bovine serum, charcoal stripped (Thermo Fisher Scientific). On day 42, depending on the experiment, iRPE cells were treated for 1 h or 24 h with the following corticosteroids treatments: aldosterone (10^−9^ M, 10^−8^ M, 10^−7^ M or 10^−6^ M), aldosterone (10^−7^ M) plus spironolactone (10^−5^ M), aldosterone (10^−7^ M) plus RU-486 (10^−5^ M), cortisol (10^−9^ M, 10^−8^ M, 10^−7^ M or 10^−6^ M), cortisol (10^−7^ M) plus spironolactone (10^−5^ M), cortisol (10^−7^ M) plus RU-486 (10^−5^ M). As corticosteroids were dissolved in ethanol (EtOH) or methanol (MeOH), control cells were treated with 0.1% EtOH or MeOH in medium.

### 4.3. Qualitative RT-PCR and RT-qPCR

Total RNA was isolated from iRPE cells using the RNeasy Mini Kit (QIAGEN, Hombrechitikon, Switzerland). For qualitative analysis, PCR was performed on cDNA samples with the KAPA Taq PCR Kit (Merck, Schaffhausen, Switzerland) and PCR products were visualized on 2% agarose gels. RT-qPCR using a LightCycler^®^ 96 (Roche Applied Science, Basel, Switzerland) and SYBR^®^ Green detection method. The following equation, which is a function of PCR efficiency and quantification cycles of both the target and the reference gene, was used for final ratio calculation. Ratio = E_R_^CqR^ / E_T_^CqT^, where E_R_, Cq_R_, E_T_ and Cq_T_ are the amplification efficiency of the reference gene, the quantification cycle of the reference gene, the amplification efficiency of the target gene and the quantification cycle of the target gene, respectively. Standard curve was set up with five dilution steps and mixed cDNA pooled from all condition samples in order to determine the amplification efficiency of target and reference genes. The PCR efficiency was calculated using the formula E = 10^−1/slope^. All primers’ amplification efficiencies ranged between 1.90 and 2.10. Housekeeping genes *GAPDH* (human cDNA samples), *β-Actin* (mouse cDNA samples) and *Rpl8* (rat cDNA samples) served as reference genes for normalization. Relative mRNA expression values for each gene of interest were set as 1 in the control and were represented as fold changes in experimental conditions. Gene-specific primers used for qualitative RT-PCR and RT-qPCR analysis are listed in Appendix A.

### 4.4. RNA Sequencing

Total RNA samples extracted from iRPE were sequenced at the iGenSeq transcriptomic platform of the Brain and Spine Institute (ICM, Paris, France). RNA quality was checked by capillary electrophoresis (Agilent 2100 Bioanalyzer system, Les Ulis, France), and RNA with integrity numbers (RIN) ranging from 7.8 to 8.2 was accepted for library generation. Quality of raw data was evaluated with FastQC. Libraries were prepared with Roche KAPA mRNA HyperPrep kit and sequenced with the Illumina NextSeq 500 Sequencing system using NextSeq 500 High Output Kit v2 (150 cycles), 400 million of reads, 50Gbases. STAR v2.5.3a was used to align reads on reference genome hg19 using standard options. Quantification of gene and isoform abundances was performed with RSEM 1.2.28, prior to normalization on library size with edgeR bioconductor package. Finally, differential analysis was also conducted with edgeR. Multiple hypothesis adjusted *p*-values were calculated with the Benjamini–Hochberg procedure to control FDR.

### 4.5. Western Blot Analysis

For the analysis of endogenous MR and GR protein expression in the cytoplasmic and nuclear fractions, iRPE cells were seeded at P3 on Matrigel-coated 6-well cell culture plates and grown for 42 days. Following aldosterone or cortisol (10^−6^ M) 1 h treatment, cells were placed on ice and washed twice with cold PBS. Total cytoplasmic and nuclear proteins were extracted using the NE-PER Nuclear and Cytoplasmic Extraction Reagents (Thermo Fisher Scientific) according to the manufacturer’s instructions. For PTGER2 and S100A2 endogenous protein expression analysis, iRPE cells were plated and grown at P3 in Matrigel-coated transwell cell culture inserts (Merck) for 42 days. Following experimental 24 h treatments, the RIPA lysis and extraction buffer (Tris HCL at pH 8 (50 mM); NaCl (150 mM); 1% NP-40; 0.5% sodium deoxycholate; 0.1% sodium dodecyl sulfate (SDS)), complemented with protease (Merck) and phosphatase (Merck) inhibitor cocktails, was used to recover total proteins. Samples were homogenized, centrifuged at 14′000 rpm for 30 min at 4 °C, and supernatants were collected. Protein concentration was measured (Pierce BCA Protein Assay Kit, Thermo Fisher Scientific) before dilution and denaturation. Protein samples (30–40 µg), along with a molecular weight marker were separated by SDS-PAGE and then transferred onto a nitrocellulose or a PVDF membrane (BIO-RAD, Cressier, Switzerland). Membranes were blocked 1 h at room temperature (RT) in 1X blocking buffer (Tris base (20 mM); NaCl (137 mM); Milli-Q H_2_O; pH 7.6; 0.1% Tween 20; 1–5% nonfat dry milk or 2% bovine serum albumin). Blots were incubated with primary antibodies overnight at 4 °C, washed three times for 5 min, incubated for 1 h at RT with secondary HRP-conjugated antibodies and washed again three times for 5 min before being developed using chemiluminescent HRP substrate detection kits (Advansta, Luzern, Switzerland). Bands were revealed (Azure Biosystems, Baden, Switzerland) and quantified using the ImageJ software (National Institutes of Health, University of Wisconsin, WI, USA). List of primary and secondary antibodies and dilutions used for Western blot analysis is available in Appendix A).

### 4.6. Immunocytochemistry on iRPE Cells

iRPE cells were cultured on coverslips or 12 mm diameter transwells (Sigma-Aldrich, Schaffhausen, Switzerland) and coated with MRF (at a 1:30 dilution) in RDMsA for 42 days before processing for immunostaining. Briefly, cells were fixed in 4% paraformaldehyde for 10–20 min at RT, rinsed with PBS and blocked for 3–5 h in 5% fetal bovine serum + 5% normal goat serum + 0.1% Triton X-100 in PBS 1X. Primary antibodies (Appendix A) were incubated overnight at 4 °C in blocking solution. Primary antibodies were washed with PBS before secondary antibody (Appendix A) incubation for 1 h at RT in PBS. After washing steps with PBS, transwell membranes were cut and mounted in Mowiol. Z-stacks were acquired using a LSM700 confocal microscope (Zeiss, Feldbach, Switzerland). For immunocytochemistry on cell sections, after cell fixation, transwell membranes were embedded on OCT compound and 7 µm sections were performed. Immunostainings were performed as described above. Images were acquired with a BX60 microscope from Olympus equipped with a DP72 camera or a Leica DM6B microscope equipped with a DFC9000GT camera and with confocal microscope.

### 4.7. Corticosteroid Profiling in Human Ocular and Cell Culture Media

The profile of corticosteroid hormones was established using a highly sensitive and specific liquid chromatographic method coupled with tandem mass spectrometric detection (LC-MS/MS) that allows simultaneous quantification of steroids, targeting the mineralocorticoid (progesterone,11-deoxycorticosterone, corticosterone, 18-hydroxycorticosterone and aldosterone) and the glucocorticoid (17-hydroxyprogesterone, 11-deoxycortisol, cortisol (F) and cortisone (E)) pathways as previously described (Travers, S et al. Multiplexed steroid profiling of gluco- and mineralocorticoids pathways using a liquid chromatography tandem mass spectrometry method, J Steroid BiochemMolBiol 2017). HSD enzymes interconvert the active cortisol and inactive cortisone. By determining the product-to-substrate ratio (E/F), we compared the HSD activities between different samples.

### 4.8. Transgenic Mice

P1.hMR mice, in which the transcription of hMR is directed by the proximal P1 promoter, exhibit a widespread and relatively strong transgene expression [65]. Wild-type animal littermates with normal endogenous MR expression and no human MR transgene expression were used as controls. Mice were housed in the animal facility under pathogen-free conditions and in a temperature- (23 ± 1 °C) and humidity-controlled (60%) room with an automatic 12 h light/12 h dark cycle. Laboratory chow and tap water were supplied ad libitum. Animal maintenance and experimental procedures were in accordance with the European Communities Council Directives 86/609/EEC and the French national regulations and approved by the local ethical committee “Comité d’éthique en experimentation animal CAPSUD” under the identification number 13837/2018022715591505 V4.

### 4.9. RT-qPCR of RPE–Choroid Complex

RPE–choroid complexes were dissected from enucleated eyes, snap-frozen in liquid nitrogen and stored at −80 °C until use. Total RNA was isolated using the RNeasy Mini Kit (QIAGEN). First-strand complementary DNA was synthesized using random primers (Thermo Fisher Scientific) and SuperScript II reverse transcriptase (Thermo Fisher Scientific). Transcript levels of mouse *Nr3c2*, human *NR3C2* and mouse *Nr3c1* were analyzed by quantitative PCR performed in CFX384 Touch Real-Time PCR Detection System with SYBR Green detection. *β-Actin* was used as housekeeping gene. Delta CT threshold calculation was used for relative quantification of results. The sequences of primers were: mouse *Nr3c2*, forward 5′-ATG GAA ACC ACA CGG TGA CCT-3′, reverse 5′-GCC TCA TCT CCA CAC ACC AAG-3′; human *NR3C2*, forward 5′-CCC TCT GAA CAT GAC ATC TTC G-3′, reverse 5′-CTG GAG CCT CGA TTT TCA AC-3′; mouse *Nr3c1*, forward 5′-TTC GCA GGC CGC TCA GTG TT-3′, reverse 5′-TTG GGA GGT GGT CCC GTT GCT-3′; *β-Actin*, forward 5′-AAG TAC CCC ATT GAA CAT GGC A-3′, reverse 5′-CAT CTT TTC ACG GTT GGC CTT A-3′.

### 4.10. In Vivo Retinal Morphology

Retinal morphology was assessed in vivo under anesthesia (ketamine 100 mg/kg, xylazine 10 mg/kg) using MICRON III image-guided optical coherence tomography (OCT, Phoenix Research Labs, Pleasanton, CA, USA). Pupils were dilated with drops of MYDRIATICUM (2 mg/0.4 mL, VIDAL, Issy les Moulineaux, France). OCT scan location and direction were visualized on a bright-field retinal image, and OCT real-time images of the retinal cross sections passing through optic nerve were taken with MICRON OCT software.

### 4.11. Retinal Histology

Enucleated eyes from 9-months-old P1hMR transgenic and wild-type mice were fixed with 4% paraformaldehyde (PFA) and 0.5% glutaraldehyde for 2 h, dehydrated in a graded alcohol series and embedded in historesin (Leica, Heidelberg, Germany). Sections of 5 µm were obtained using a Leica Jung RM2055 microtome and stained with 1% toluidine blue. Retinal morphology was observed in bright field using Olympus BX51 microscope (Olympus, Rungis, France). For quantification, cross sections at the level of optic nerve head were selected. Serial photographs were taken with 40× magnification starting from the region adjacent to the optic nerve to the periphery. Measurements of area of choroid and photoreceptor segments were performed on 10 photographs/eye, 5 on each side of optic nerve using ImageJ. Photoreceptor segments were measured from the outer side of the outer nuclear layer to the apices of the RPE cells, and the choroid was measured from the outer side of the Bruch’s membrane to the lamina fusca. Thickness of RPE cells was measured from the RPE villosities to the basal membrane every 200µm nasal and temporal to the optic nerve on at least 10 photographs/eye (*n* = 8 eyes per group). Average choroidal and photoreceptor segment area per photograph/eye and average RPE thickness/eye were used for comparison.

### 4.12. Immunofluorescence of Albumin

Enucleated eyes were fixed in 4% PFA for 2 h, then snap-frozen in Tissue-Tek OCT compound (Bayer Diagnostics, Puteaux, France). Cryosections (10 µm) were collected on slides. After permeabilization with 0.1% Triton X100, sections were incubated with FITC-coupled goat anti-mouse albumin antibody (1:100, Bethyl, Montgomery, TX, USA) for 1 h. Cell nuclei were counter-stained with 4′,6-Diamidino-2-Phenyl-Indole (DAPI, 1:5000, Sigma-Aldrich, Saint Quentin Fallavier, France). Images were taken using an Olympus BX51 fluorescence microscope.

### 4.13. Statistics

Quantitative data were expressed as mean ± SEM. Statistical analysis was conducted using the GraphPad Prism 5 program (GraphPad Software, San Diego, CA, USA). The unpaired *t*-test with Welch’s correction was used to compare two groups. The Kruskal–Wallis and Dunn’s test were used to compare more than two groups. *p* < 0.05 was considered significant.

## Data Availability

All data are available upon reasonable request. Raw RNA seq data will be made available on Gene Expression Omnibus GSE172478 (https://www.ncbi.nlm.nih.gov/geo/query/acc.cgi?acc=GSE172478, accessed on 20 July 2021).

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
