# Peer review of "Pathogenic Effects of Mineralocorticoid Pathway Activation in Retinal Pigment Epithelium"

_ijms, 2021, doi:10.3390/ijms22179618_

Round 1

Reviewer 1 Report

In 'Pathogenic effects of mineralocorticoid pathway activation in 2 retinal pigment epithelium' Canonica et al. used iRPE cells and transgenic mice that over-express the human mineralocorticoid receptor to test the effects of aldosterone, cortisol or cortisol + RU486 on RPE cells. The manuscript embodies a wealth of results obtained with a combination of methods including RNAseq, ICC, Western blot etc. The authors found that a subset of genes regulated by aldosterone and by cortisol + RU-486, and not by cortisol alone mimics an imbalance toward MR activation. MR pathway activation in RPE cells regulates genes involved in barrier function, extracellular matrix, neural regulations and epithelial differentiation, which could contribute to retinal pathology.

There are a few issues with the format and language throughout the text, punctuations, grammar, typos etc.

Overall this is a nice body of work that is well worth publishing.

Author Response

We thank the reviewer for his positive evaluation of our work.

We have edited the document and tried to correct as much errors as possible.

We hope the paper has been improved. 

We are ready to send it to a professional editor if needed

Best regards

Reviewer 2 Report

The paper entitled “Pathogenic effects of mineralocorticoid pathway activation in retinal pigment epithelium” by Canonica et al., deals with the study of the transcriptional regulations induced by glucocorticoid receptor (GR) and mineralocorticoid receptor (MR) activation in retinal pigment epithelial cells (RPE)

Although the paper addresses an issue of interest in the field, the authors may wish to consider the following prior to publication.

Introduction Section: the authors should mention the concept of neurosteroids and the importance on RPE. Please add the following pioneer studies on this matter (it will be useful for the reader that is not familiar with the specific field): Neuroreport. 2005 Aug 1;16(11):1203-7. doi: 10.1097/00001756-200508010-00014; Eur J Pharmacol. 2004 Sep 13;498(1-3):111-4. doi: 10.1016/j.ejphar.2004.06.0; J Neurochem. 1994 Jul;63(1):86-96. doi: 10.1046/j.1471-4159.1994.63010086.x.

The figures are too crowded. Please consider the possibility to re-arrange the panels and put some figure as supplemental material.

Please check the English style

Please check the reference section (see Journal Guidelines)

Author Response

We thank the reviewer for his positive evaluation of our manuscript.

As requested, we have changed the introduction to mention previous work on the role of neurosteroids in the retina and RPE. We thank the reviewer for this comment as these papers are indeed of major interest in the context of our work.

The introduction was modified as follows:

The role of corticoids in retinal homeostasis and in diseases remains incompletely understood and underestimated. In the rat retina,  neurosteroids produced by steroidogenesis in rat retinal ganglion cells [8] protected retinal neurons against ischemia through binding to sigma-1 receptor [9], that regulates calcium, ion channels and oxidative stress [10]. In a human RPE cell line (ARPE-19), 17β-estradiol and dehydroepiandrosterone-sulfate protected against oxidative stress-induced DNA damages[11], demonstrating the crucial role of neurosteroids in retina and RPE maintenance. In the same RPE cell line, key enzymes participating in steroids synthesis were identified and transformation of progesterone into cortisol was found in culture media, suggesting that a equivalent of hypothalamic–pituitary–adrenal axis in adult retinal pigment epithelium may exist.[12] Regarding the figures, it is worth mentioning that we have already put in supplementary figures an important part of the work. In the document that was provided for review, we had added in the main text 4 supplementary figures. Thus, in the main document, only 9 figures were included. We think that this information is the minimal required to follow the text and understand the results.  To reduce the amount of information in each figure, we have divided figure 2, 5 and 9 into two separate figures. The new figures are now provided in the revised document. We have edited the document to correct mistakes as much as we could. We are ready to send the paper to a professional editor if needed. The reference format has been changed. 
